# Combinatorial Optimization with Policy Adaptation using Latent Space Search

**Felix Chalumeau***
InstaDeep
f.chalumeau@instadeep.com

**Shikha Surana***
InstaDeep
s.surana@instadeep.com

**Clément Bonnet**
InstaDeep
c.bonnet@instadeep.com

**Nathan Grinsztajn**
InstaDeep
n.grinsztajn@instadeep.com

**Arnu Pretorius**
InstaDeep
a.pretorius@instadeep.com

**Alexandre Laterre**
InstaDeep
a.laterre@instadeep.com

**Thomas D. Barrett**
InstaDeep
t.barrett@instadeep.com

## Abstract

Combinatorial Optimization underpins many real-world applications and yet, designing performant algorithms to solve these complex, typically NP-hard, problems remains a significant research challenge. Reinforcement Learning (RL) provides a versatile framework for designing heuristics across a broad spectrum of problem domains. However, despite notable progress, RL has not yet supplanted industrial solvers as the go-to solution. Current approaches emphasize pre-training heuristics that construct solutions but often rely on search procedures with limited variance, such as stochastically sampling numerous solutions from a single policy or employing computationally expensive fine-tuning of the policy on individual problem instances. Building on the intuition that performant search at inference time should be anticipated during pre-training, we propose COMPASS, a novel RL approach that parameterizes a distribution of diverse and specialized policies conditioned on a continuous latent space. We evaluate COMPASS across three canonical problems - Travelling Salesman, Capacitated Vehicle Routing, and Job-Shop Scheduling - and demonstrate that our search strategy (i) outperforms state-of-the-art approaches in 9 out of 11 standard benchmarking tasks and (ii) generalizes better, surpassing all other approaches on a set of 18 procedurally transformed instance distributions.

## 1 Introduction

Combinatorial Optimization (CO) has a wide range of real-world applications, from transportation (Contardo et al., 2012) and logistics (Laterre et al., 2018), to energy (Froger et al., 2016). Solving a CO problem consists of finding an ordering, labelling or subset of elements from a finite, discrete set that maximizes (or minimizes) a given objective function. As the number of feasible solutions typically grows exponentially with the problem size, CO problems are challenging (often NP-hard) to solve. As such, significant work goes into designing problem-specific heuristic approaches that,

---

*Equal contribution

37th Conference on Neural Information Processing Systems (NeurIPS 2023).

whilst not guaranteeing the optimal answer, can often work well in practice. Reinforcement Learning (RL) offers a domain-agnostic framework to learn heuristics and has been successfully applied across a range of CO tasks (Vinyals et al., 2015; Deudon et al., 2018; Mazyavkina et al., 2021).

Concretely, leading RL methods typically train a policy to incrementally *construct* a solution one element at a time. However, whilst most efforts have focused on improving the one-shot quality of these construction heuristics (Kool et al., 2019; Kwon et al., 2020; Grinsztajn et al., 2022), it intuitively appears impractical to reliably produce the optimal solution to NP-hard problems within a single construction attempt. Consequently, competitive performance has to rely on combining a pre-trained policy with an additional search procedure. Nevertheless, this crucial aspect is often implemented using simple procedures such as stochastic sampling (Kool et al., 2019; Kwon et al., 2020; Grinsztajn et al., 2022), beam search (Steinbiss et al., 1994) or Monte Carlo Tree Search (MCTS) (Browne et al., 2012). An alternative approach, representing the current state-of-the-art for search-based RL (Bello et al., 2016; Hottung et al., 2022), is to actively re-train the heuristic on each new problem instance; however, this comes with clear computational and practical limitations. Strikingly, neither of these approaches pre-trains the policy in a way that could enable a fast and efficient inference time search: rather current approaches typically completely decouple both. The absence of an efficient search strategy is even more detrimental when the test instances are out of the distribution (OOD) used to train the policy, as this may cause a large difference between the learned policy and the policy leading to the optimal solution.

In this work, we aim to overcome the current limitations of search strategies used in RL when applied to CO problems. Our approach is to learn a latent space of diverse policies that can be explored at inference time in order to find the best-performing strategies for a given instance. This updates the current paradigm by enabling sampling from a policy space at inference time rather than constantly sampling the same policy (or set of policies) with stochasticity. We introduce **COMPASS** – **COM**binatorial optimization with **P**olicy **A**daptation using Latent **S**pace **S**earch. COMPASS effectively creates an infinite set of diverse solvers by using a single conditioned policy and sampling the conditions from a continuous latent space. The training process encourages subareas of the latent space to specialize to sub-distributions of instances and this diversity is used at inference time to solve newly encountered instances.

We evaluate COMPASS on three popular CO problems: Travelling Salesman Problem (TSP), Capacitated Vehicle Routing Problem (CVRP) and Job-Shop Scheduling Problem (JSSP). After training on a distribution of fixed-sized instances for each problem, we evaluate our method on both in- and out-of-distribution test sets. We find that simple search strategies requiring no re-training provide both rapid and sustained improvement of the instance-specific policy, with COMPASS establishing a new state-of-the-art across all problems in this setting. Thanks to the diversity provided by its latent space, COMPASS achieves high performance even without a search budget and achieves comparable or better results than current leading few-shot methods.

Concretely, our work makes the following contributions: **(i)** We introduce COMPASS which leverages a latent space of diverse and specialized policies to effectively solve CO problems. **(ii)** We show that COMPASS allows for the efficient adaptation of instance-specific policies without re-training or sacrificing zero-shot performance. **(iii)** Experimentally, our approach is found to represent a new state-of-the-art for RL-based CO methods across all our considered problem types, achieving superior performance in 27 out of 29 tasks. **(iv)** We release fast and performant implementations of our method and its main competitors, written in JAX. We also provide all of our test sets including our procedurally transformed problem instances for easier comparison in future work.

## 2   Related work

**Construction methods for CO**   Construction approaches in RL for CO incrementally build a solution by selecting one element at a time. After Hopfield and Tank (1985) first applied neural networks to TSP, Bello et al. (2016) extended these efforts by proposing to learn heuristics with RL using a Pointer Network (Vinyals et al., 2015) combined with an actor-critic framework. This approach was extended by Deudon et al. (2018) who added an attention-based city encoder, which was subsequently further extended by Kool et al. (2019) to use a general transformer architecture (Vaswani et al., 2017). The transformer has since become the standard model for a range of CO problems and is also used in this work. Kim et al. (2022) builds on Kool et al. (2019) by leveraging

symmetries of routing problems during training. Even though the majority of these construction approaches have focused on routing problems, numerous works have also tackled other classes of CO problems, especially on graphs, like Maximum Cut (Dai et al., 2017; Barrett et al., 2020), or Job Shop Scheduling Problem (JSSP), for which Zhang et al. (2020) proposed a Graph Neural Network (GNN) approach. A broader scope of (non-construction) approaches can be found in Appendix K.

**Improving solutions at inference time** As it is unlikely that the first solution generated by a construction heuristic is optimal, a popular approach consists in sampling various trajectories during inference for the same CO problem. POMO (Kwon et al., 2020) uses one policy rolled out on several versions of the same problem, while considering different starting points or symmetries, to create diverse trajectories and select the best one. Choo et al. (2022) proposes an efficient search guided by simulations, but cannot take advantage of a large inference budget by itself. EAS (Hottung et al., 2022) adds on POMO by fine-tuning a subset of the model parameters at inference time using gradient descent. However, the new solutions are biased toward the underlying pre-trained policy and can easily be stuck in local optima. Instead, MDAM (Xin et al., 2021) and Poppy (Grinsztajn et al., 2022) employ a population of agents, all of which are simultaneously rolled out at inference time. MDAM trains these policies to select different initial actions, whereas Poppy utilizes a loss function designed to specialize each policy on specific subsets of the problem distribution. Despite demonstrating promising performance, these approaches are constrained by the number of policies used during training, which remains fixed. Such a limitation quickly diminishes the benefits of additional solution candidates sampled from the population. Our method COMPASS uses the same loss as Poppy, but, unlike their approach and that of MDAM, COMPASS is not bound to a specific number of specialized policies. Moreover, its latent space makes it possible to add additional search mechanisms over the policy space, ensuring better solutions over time. CVAE-Opt (Hottung et al., 2021), akin to our method, uses a latent space for solving routing problems, however, it has several differences. First, COMPASS is trained end-to-end with RL, hence does not necessitate pre-solved instances. Second, CVAE-Opt requires training an additional recurrent encoder for (instance, solution) pairs, whereas COMPASS uses the latent space to encode a distribution of complementary policies and can be easily applied to pre-train models. Overall, COMPASS significantly outperforms CVAE-Opt while having shorter runtime.

## 3 Methods

### 3.1 Preliminaries

**Formulation** The goal of a CO problem is to find the optimal labeling of a set of discrete variables that satisfies the problem's constraints. In RL, a CO problem can be formulated as a Markov Decision Process (MDP) defined by $M = (S, A, R, T, \gamma, H)$. This includes the state space $S$ with states $s_i \in S$, action space $A$ with actions $a_i \in A$, reward function $R : S \times A \to R$, transition function $T(s_{i+1}|s_i, a_i)$, discount factor $\gamma \in [0, 1]$, and horizon $H$ which denotes the episode duration. The state of a problem instance is represented as the (partial) trajectory or set of actions taken in the instance, and the next state $s_{t+1}$ is determined by applying the chosen action $a_t$ to the current state $s_t$. An agent is introduced in the MDP to interact with the CO problem and find solutions by learning a policy $\pi : S \to A$. The policy is trained to maximize the expected sum of discounted rewards to find the optimal solution, and this is formalized as the following learning objective: $\pi^* = \underset{\pi}{\arg\max} \ \mathbb{E}[\sum_{t=0}^{H} \gamma^t R(s_t, a_t)]$.

### 3.2 COMPASS

Recall our intuition that no single policy will reliably be able to solve all instances of an NP-hard CO problem in a single inference pass. Two primary approaches to address this are the inclusion of inference time search and the deployment of a diverse set of policies to increase the chance of a near-optimal strategy being deployed. This work aims to unify and extend these approaches by training an infinitely large set of diverse and specialized policies that can subsequently be searched at inference time.

To achieve this, we propose that a single set of policy parameters condition not just on the current observation, but also on samples drawn from a continuous latent space. The training objective

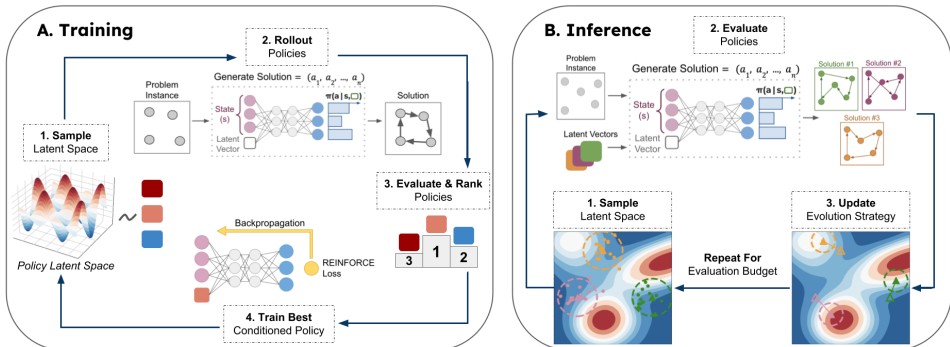

Figure 1: Our method COMPASS is composed of the following two phases. A. Training - the latent space is sampled to generate vectors that the policy can condition upon. The conditioned policies are then evaluated and only the best one is trained to create specialization within the latent space. B. Inference - at inference time the latent space is searched through an evolution strategy to exploit regions with high-performing policies for each instance.

then encourages this latent space of policies to be diverse (generate a wide range of behaviors) and specialized (these behaviors are optimized for different types of problem instances from the training distribution). This latent space can then be efficiently searched during inference to find the most performant policy for a given problem instance. In this section, we describe in detail the realization of this approach, which we call COMPASS (**COM**binatorial optimization with **P**olicy **A**daptation using Latent **S**pace **S**earch). In Fig. 1, we provide an illustrated overview of COMPASS.

Our approach offers several key advantages over traditional techniques. Compared to methods that directly train multiple, uniquely parameterized policies (Xin et al., 2021; Grinsztajn et al., 2022), training a single conditional policy can, in principle, provide a continuous distribution of an infinite number of policies. Moreover, our approach mitigates the significant training and memory overheads associated with training a population of agents. Compared to methods that rely on brute-force sampling (Kool et al., 2019; Kwon et al., 2020; Grinsztajn et al., 2022) or expensive fine-tuning (Hottung et al., 2022), our training process produces a structured latent space (where similar policies are found near to each other) that permits principled search during inference.

**Latent space** The latent space defines the set of policies that our model can condition itself upon. Importantly, we do not learn the distribution of this space, but rather select a prior distribution over the space from which we sample during training. In practice, we use a latent space with 16 dimensions bounded between -1 and 1, and use a uniform sampling prior.

**Architecture** COMPASS is agnostic to the network architecture used, so long as the resulting policy is, in some way, conditioned on the vector sampled from the latent space. This can be achieved in numerous ways, from directly concatenating the vector to the input observation to conditioning keys, queries, and values in the self-attention models commonly used for CO. We refer to Appendix D for further details about the architectures used in this work and how the latent vector is used to condition them. Whilst it is possible to train COMPASS from scratch, we found that it was simple and efficient to adapt pre-trained single-policy models. To do this, we zero-initialize any additional weights corresponding to the sample latent vector such that it has no impact at the start of training. In practice, we adapt a single-agent architecture designed for few-shot inference in all of our problem settings; POMO (Kwon et al., 2020) for TSP and CVRP, and a similar architecture taken from Jumanji (Bonnet et al., 2023) for JSSP (we also considered the current SOTA model L2D (Zhang et al., 2020), however, we found that the model from Jumanji already outperformed this approach). Full network details can be found in Appendices D.1 (TSP & CVRP) and D.2 (JSSP).

**Training** The training procedure aims to specialize subareas of the latent space to sub-distributions of problems by training the policy solely on latent vectors that achieve the best performance on a given problem. At each training step, we uniformly sample a set of $N$ vectors from the latent space and condition the policy on each vector resulting in $N$ conditioned policies. After evaluating each policy

on the problem instance, we train the best policy (i.e. the policy conditioned on the best-performing latent vector) on the instance. The model is updated using the gradient of our objective as given by

$$\nabla_\theta J_{\text{compass}} = \mathbb{E}_{\rho \sim \mathcal{D}} \mathbb{E}_{z_1, \ldots, z_N \sim \mathcal{P}_z} \mathbb{E}_{\tau_i \sim \pi_\theta(\cdot | z_i)} [\nabla_\theta \log \pi_\theta(\tau_{i^\star} | z_{i^\star}) R_{i^\star} - \mathcal{B}_{\rho, \theta}], \tag{1}$$

where $\mathcal{D}$ is the data distribution, $\mathcal{P}_z$ the latent space, $z_i$ a latent vector, $\pi_\theta$ the conditioned policy, $\tau_i$ the trajectory generated by policy $\pi_\theta$ conditioned on vector $z_i$ and has the corresponding reward $R_i$, $i^\star$ is the index of the best performing latent vector (in the sampled set) and is expressed as $i^\star = \arg \max_{i \in [1, N]} R(\tau_i)$, and lastly, $\mathcal{B}_{\rho, \theta}$ is the baseline, inspired by Kwon et al. (2020). Full details of the algorithmic procedure can be found in Appendix F. Notably, our work is the first to create a specialized and diverse set of policies represented by a continuous latent space by only training the best-performing vector for each problem instance.

A key training hyperparameter is the number of condition vectors sampled during evaluation. More conditioned policies results in an increased certainty that the best-performing vector in the sampled set of conditions is the best-performing vector in the latent space. Therefore, increasing the number of sampled conditions increases the likelihood of training the true best latent vector for the given problem instance, rather than a potentially suboptimal vector. More details (including training times and environment steps) are reported in Appendix F.

**Inference-time search**    Given the latent space of diverse, specialized policies obtained by training COMPASS, at inference time, we apply a principled search procedure to find the most performant strategies. Our desired properties for a search procedure are that it should be simple, capable of rapid adaptation and robust to local optima. As such, evolutionary strategies are an appropriate approach. Specifically, we use Covariance Matrix Adaptation (CMA-ES, (Hansen and Ostermeier, 2001)). CMA-ES uses a multivariate normal distribution to sample vectors and iteratively updates the distribution's mean to increase the expected performance of sampled vectors (i.e. the quality of the solution found by the policy corresponding to each vector). The covariance is also adapted over time, either for exploration (high values, broad sampling) or exploitation (small values, focused sampling).

For a given problem instance, there may be multiple high-performance policies, therefore we use several independent CMA-ES components in parallel. To ensure that those components explore distinct areas of the space (or at least, take different paths), we compute a Voronoi Tesselation (Du et al., 1999) of the latent space and use the corresponding centroids to initialize the means of the CMA-ES components. This method proves to be robust, easy to tune, and fast, and requires low memory and computation budget, making it the perfect candidate for efficient adaptation at inference time. In our experimental section (4.3), we present an analysis of our latent space and how it is explored by CMA-ES. Details and considered alternatives can be found in Appendix E.4.

## 4    Experiments

We evaluate our method on three problems – Travelling Salesman (TSP), Capacitated Vehicle Routing (CVRP), and Job Shop Scheduling (JSSP) – widely used to assess RL-based methods for CO (Deudon et al., 2018; Kool et al., 2019; Grinsztajn et al., 2022; Hottung et al., 2022). In Section 4.1, we evaluate COMPASS in the standard setting used by other methods from the literature and report results on each problem type. In Section 4.2, we assess the robustness of methods by evaluating them on instances of TSP and CVRP that are procedurally transformed using the approach developed by Bossek et al. (2019). In Section 4.3, we analyze the methods' search strategies; in particular, we provide insights about COMPASS' latent space and how it is navigated by CMA-ES at inference time. Figure 2 provides a radar plot overview of our aggregated experimental results across six performance categories of interest: (1) in distribution instances, OOD instances with different levels of distribution shift (2) small, (3) medium and (4) large, (5) large instance sizes as well as (6) few-shot performance. Our results highlight the strengths and weaknesses of each approach and in particular, the versatility and superiority of COMPASS.

**Baselines**    We compare COMPASS to a suite of leading RL methods and industrial solvers. Across all problems we provide baselines for EAS (Hottung et al., 2022); the current SOTA active-search RL method that fine-tunes the policy on each problem instance, and Poppy (Grinsztajn et al., 2022); the current SOTA active-search RL method that stochastically samples from a fixed population of

pre-trained solvers. For routing problems (TSP and CVRP), we also provide results for POMO (Kwon et al., 2020); the leading single-agent, one-shot architecture on which EAS and Poppy are built, and LKH (Helsgaun, 2017); a leading industrial solver. We also report results of TSP-specific industrial solver Concorde (Applegate et al., 2006). For JSSP, we provide results for L2D (Zhang et al., 2020); the leading single-agent, one-shot architecture. We also provide results for the attention-based model proposed in Jumanji (Bonnet et al., 2023) that proved to outperform L2D. Finally, we report the results of Google OR-Tools (Perron and Furnon, 2019); the reference industrial solver for JSSP.

**Training**    As our method is capable of adopting initial parameters from a pre-trained model, we re-use publicly available checkpoints of POMO (details in Appendix H) as the starting point for COMPASS on TSP and CVRP. For JSSP, we found attention-based model from Bonnet et al. (2023) outperforms L2D and hence choose it to be the reference single-agent architecture. We train the model and use the same trained checkpoints for all methods. We then train COMPASS until convergence on the same training distribution as that used to train the initial checkpoint. For TSP and CVRP these are problem instances with 100 locations uniformly sampled within a unit square. For JSSP, we use the same training distribution used in EAS, which is an instance with 10 jobs and machines, and a maximum possible duration of 98. A single set of hyperparameters is used across all problems, with full training details provided in Appendix G.

**Inference**    When evaluating active-search performance, each method is given a fixed budget of 1600 attempts – similar to Hottung et al. (2022); Grinsztajn et al. (2022) –, where each attempt consists of one trajectory per possible starting point. This approach is used to enable direct comparison to POMO and EAS which use rollouts from all starting points at each step. For the main results on TSP and CVRP, we do not use the "augmentation trick"; where the same problem is solved multiple times by rotating the coordinate frame to make it appear different and thus generate additional diverse trajectories. This trick was used in a few baselines from prior work, however, we refrain from using it in the main results of this work for two reasons: (1) it is a domain-specific trick mainly applicable to routing problems and (2) it significantly increases the required computational budget. We nevertheless provide some results in both settings to ease comparison with previous work. Overall, the trajectory budget is exactly the same as the one used in Grinsztajn et al. (2022); Hottung et al. (2022). Note that expressing the budget in terms of trajectories gives an advantage to EAS, which uses more time, memory, and computation due to the backpropagations used to update the policy during the search.

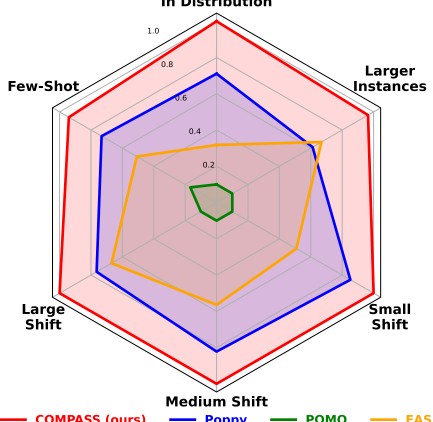

**In Distribution**    Instances drawn from the training distribution (see first cols. in Tables 1a to 1c).

**Larger Instances**    Instance with a larger size than the training data. See *generalization* in Tables 1a to 1c.

**Small/Medium/Large Shift**    Instances that have been procedurally mutated to be out of the training distribution. Detailed results in Fig. 3.

**Few-Shot**    The standard benchmark with a lower budget (typically 10% of the usual). Numerical results in Appendix I.

Figure 2: Performance of COMPASS and the main baselines aggregated across several tasks over three problems (TSP, CVRP, and JSSP). For each task (problem type, instance size, mutation power), we normalize values between 0 and 1 (corresponding to the worst and best performing RL method, respectively). Hence, all tasks have the same impact on the aggregated metrics. COMPASS surpasses the baselines on all of them, showing its versatility for all types of tasks and in particular, its generalization capacity.

Table 1: Results of COMPASS against the baseline algorithms for (a) TSP, (b) CVRP, and (c) JSSP problems. The methods are evaluated on instances from training distribution as well as on larger instance sizes to test generalization.

(a) TSP

| | Training distr. | | | Generalization | | | | | | | | |
| | $n = 100$ | | | $n = 125$ | | | $n = 150$ | | | $n = 200$ | | |
| Method | Obj. | Gap | Time | Obj. | Gap | Time | Obj. | Gap | Time | Obj. | Gap | Time |
|---|---|---|---|---|---|---|---|---|---|---|---|---|
| Concorde | 7.765 | 0.000% | 82M | 8.583 | 0.000% | 12M | 9.346 | 0.000% | 17M | 10.687 | 0.000% | 31M |
| LKH3 | 7.765 | 0.000% | 8H | 8.583 | 0.000% | 73M | 9.346 | 0.000% | 99M | 10.687 | 0.000% | 3H |
| POMO (greedy) | 7.796 | 0.404% | 37S | 8.635 | 0.607% | 6S | 9.440 | 1.001% | 10S | 10.933 | 2.300% | 21S |
| POMO (sampling) | 7.779 | 0.185% | 2H | 8.609 | 0.299% | 20M | 9.401 | 0.585% | 32M | 10.956 | 2.513% | 70M |
| Poppy 16 | 7.766 | 0.013% | 2H | 8.587 | 0.050% | 20M | 9.359 | 0.141% | 32M | 10.795 | 1.007% | 70M |
| EAS | 7.779 | 0.176% | 7H | 8.601 | 0.252% | 62M | 9.382 | 0.381% | 2H | 10.758 | 0.660% | 210M |
| **COMPASS (ours)** | **7.765** | **0.002%** | 2H | **8.586** | **0.036%** | 20M | **9.354** | **0.083%** | 32M | **10.724** | **0.348%** | 70M |

(b) CVRP

| | Training distr. | | | Generalization | | | | | | | | |
| | $n = 100$ | | | $n = 125$ | | | $n = 150$ | | | $n = 200$ | | |
| Method | Obj. | Gap | Time | Obj. | Gap | Time | Obj. | Gap | Time | Obj. | Gap | Time |
|---|---|---|---|---|---|---|---|---|---|---|---|---|
| LKH3 | 15.65 | 0.000% | - | 17.50 | 0.000% | - | 19.22 | 0.000% | - | 22.00 | 0.000% | - |
| POMO (greedy) | 15.874 | 1.430% | 2M | 17.818 | 1.818% | <1M | 19.750 | 2.757% | 1M | 23.318 | 5.992% | 2M |
| POMO (sampling) | 15.713 | 0.399% | 4H | 17.612 | 0.642% | 43M | 19.488 | 1.393% | 1H | 23.378 | 6.264% | 100M |
| Poppy 32 | 15.663 | 0.084% | 4H | 17.548 | 0.276% | 42M | 19.421 | 1.044% | 1H | 23.352 | 6.144% | 100M |
| EAS | 15.661 | 0.068% | 13H | 17.517 | 0.094% | 2H | **19.285** | **0.341%** | 4H | **22.264** | **1.120%** | 7H |
| **COMPASS (ours)** | **15.594** | **-0.361%** | 4H | **17.511** | **0.064%** | 42M | 19.313 | 0.485% | 1H | 22.462 | 2.098% | 100M |

(c) JSSP

| | Training distr. | | | Generalization | | | | | |
| | $10 \times 10$ | | | $15 \times 15$ | | | $20 \times 15$ | | |
| Method | Obj. | Gap | Time | Obj. | Gap | Time | Obj. | Gap | Time |
|---|---|---|---|---|---|---|---|---|---|
| OR-Tools | 807.6 | 0.0% | 37S | 1188.0 | 0.0% | 3H | 1345.5 | 0.0% | 80H |
| L2D (sampling) | 871.7 | 8.0% | 8H | 1378.3 | 16.0% | 25H | 1624.6 | 20.8% | 40H |
| Single | 862.1 | 6.7% | 3H | 1302.6 | 9.6% | 5H | 1503.0 | 11.7% | 8H |
| Poppy 16 | 849.7 | 5.2% | 3H | 1290.4 | 8.6% | 5H | 1495.7 | 11.2% | 8H |
| EAS | 858.4 | 6.3% | 5H | 1295.2 | 9.0% | 9H | 1498.0 | 11.3% | 11H |
| **COMPASS (ours)** | **845.5** | **4.7%** | 3H | **1282.8** | **8.0%** | 5H | **1485.6** | **10.4%** | 8H |

**Code availability** We release the code[1] used to train our method and to run all baselines. We also make our checkpoints available for all three problems, along with the datasets necessary to reproduce the results. To ensure fair comparison and extend our evaluation to new settings, we reimplemented all baselines within the same codebase. For the three problems, we used the JAX (Bradbury et al., 2018) implementations from Jumanji (Bonnet et al., 2023) to leverage hardware accelerators (e.g. TPU). Our code is optimized for TPU v3-8, which is the hardware used for our experiments.

## 4.1 Standard benchmarking on TSP, CVRP, and JSSP

We evaluate our method on benchmark sets frequently used in the literature (Kool et al., 2019; Kwon et al., 2020; Grinsztajn et al., 2022; Hottung et al., 2022). Specifically, for TSP and CVRP, we use datasets of 10 000 instances drawn from the training distribution, with the positions of 100 cities/customers uniformly sampled within the unit square, and three datasets not seen during training, each containing 1000 problem instances but with larger sizes: 125, 150 and 200, also generated from a uniform distribution over the unit square. We use the exact same datasets as in the literature.

**Results** The average performance of each method across all problem settings are presented in Table 1. We find that COMPASS demonstrates superior performance on 9 out of the 11 test sets considered. Moreover, the degree of improvement is significant across all problem types. On TSP and JSSP, COMPASS reduces the optimality gap on the training distribution by a factor of 6.5 and 1.3, respectively. On CVRP, COMPASS is the only RL method able to outperform the industrial solver LKH. Finally, COMPASS is also found to generalize well to larger problem instances unseen during training. COMPASS obtains the best solutions in all TSP and JSSP sets and is only outperformed by

---

[1]Code, checkpoints and evaluation sets are available at `https://github.com/instadeepai/compass`

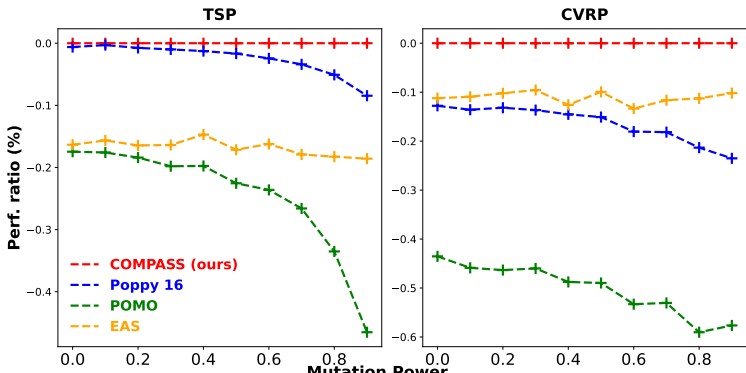

Figure 3: Relative difference between COMPASS and baselines as a function of mutation power. COMPASS outperforms the baselines on all 18 evaluation sets. Most methods have a decreasing performance ratio, showing that COMPASS generalizes better: its evolution strategy is able to find areas of its latent space that are high-performing, even on instances that are out-of-distribution.

EAS on two instance sizes of CVRP. Nevertheless, EAS is 50% slower and more computationally expensive as it requires updating an entire subset of its network's weights (see Appendix A), as opposed to simply navigating the 16-dimensional latent space of policies as is done in COMPASS.

The same benchmark is also reported with the "augmentation trick" in Table 2 for TSP and CVRP. This trick can only be used for the routing problem and is not applicable for JSSP. Interestingly, COMPASS is the only method that performs on par or better without the "augmentation trick", showing its ability to adapt and find diversity in its latent space rather than through a problem-specific trick. Other conclusions drawn above remain unchanged in this setting.

## 4.2 Robustness to generalization: solving mutated instances

To further study the generalization ability of our method, we consider the mutation operators introduced by Bossek et al. (2019) to procedurally transform instances drawn from the training distribution. By progressively increasing the power of the applied mutations we construct new datasets that are increasingly far from the training distribution whilst not modifying the overall size of the problem.

We use 9 different mutation operators (explosion, implosion, cluster, rotation, linear projection, axis projection, expansion, compression and grid). One can find an illustration of the entire set of mutations along with their mathematical definition in Appendix C. Interestingly, it enables us to evaluate the methods on instances that look closer to real-life situations. For instance, the operator that gathers nodes in a cluster can mimic a dense city surrounded by its nearby suburbs. In practice, each mutation operator is parameterized by a factor that controls the probability of mutating each node of the instance - referred to as mutation power - this factor directly impacts the shift between the training distribution and the new distribution. We use 10 values, going from 0 (no change) to 0.9 (highly mutated instances).

**Results** We plot the relative performance of the baselines compared to COMPASS in Fig. 3. A negative performance ratio indicates that a method does not provide as good of a solution as COMPASS, and we observe that this is the case for all baseline methods, at all mutation strengths, on both TSP and CVRP. Moreover, COMPASS is seen to generalize significantly better than the methods that only rely on stochastic sampling for their search, namely POMO and Poppy. This validates our intuition that adaptive policies are especially important for handling out-of-distribution data, where the optimal policy may be significantly different to that needed during pre-training. Even compared to EAS, which fine-tunes the policy to the target problem instance, we find that COMPASS maintains a significant performance gap across all mutation strengths. This result is particularly noteworthy as our approach only modifies 16 parameters (the conditioning vector sampled from our latent space), compared to EAS, which updates more than $10^4$ parameters (the embeddings of the instance's nodes).

It is interesting to note that the relative generalization performance of COMPASS compared to EAS is stronger on these mutated instances than the larger instances considered in Section 4.1. We

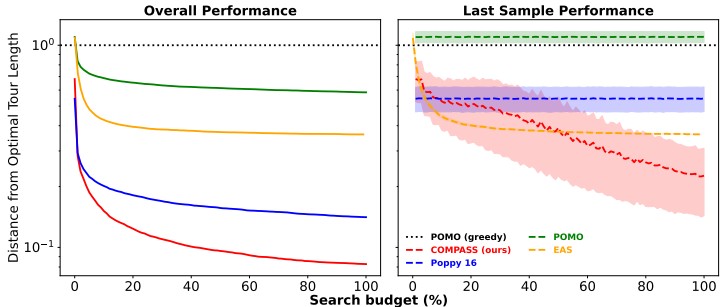

Figure 4: Evolution of the overall performance and last performance obtained by the methods during their search on TSP150 - averaged on 1000 instances. The right plot reports mean and standard deviations of the most recent shots tried by methods during the search. It illustrates how COMPASS efficiently explores its latent space to search for high-performing solutions.

hypothesize that this is because EAS actively fine-tunes the embeddings of every location in a given problem instance. Therefore, as the problem size increases, so does the number of free parameters to adapt the policy (albeit with commensurately increasing computational overhead). This suggests that further improvements to COMPASS could be possible by increasing the number of adapted parameters (i.e. the latent space dimension), however, we defer further investigation to future works.

## 4.3 Analysis of the search strategies

In this section, we analyze the structure of the latent space and the behavior of the search procedure both empirically and visually.

Figure 4 details the performance of our considered methods as a function of the overall search budget. The left panel reports the quality of the best solution found so far (i.e. the cumulative performance), whereas the right plot reports the mean and standard deviations of the latest batch of solutions (i.e. the current performance) during the search process. From this, we would highlight three main conclusions. (i) Adaptive methods (COMPASS, EAS) perform well as they are able to improve the mean performance of their solution over time, in contrast to stochastic sampling methods (Poppy, POMO). This also highlights that the latent space of COMPASS has been able to diversify and can be exploited. (ii) Highly-focused (low-variance) search does not always outperform stochastic exploration. Concretely, whilst EAS quickly adapts a policy with better average performance than Poppy (right panel), the additional variance of Poppy's multiple diverse policies means it produces better overall solutions (left panel). (iii) COMPASS is able to combine both of the pre-

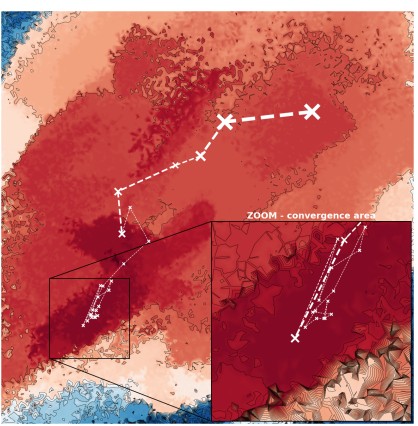

Figure 5: Contour plot of COMPASS's latent space, reflecting performance on a problem instance. White crosses show the successive means of a CMA-ES component during the search. The width of the path is proportional to the search's variance.

viously discussed aspects for a highly performant search procedure. By using an adaptive covariance mechanism as well as its multiple components to navigate several regions of the latent policy space, it focuses its search on promising strategies (better average performance) whilst maintaining a broad beam (higher variance).

To better understand how COMPASS's latent space is structured and explored, Fig. 5 presents the trajectory of a single CMA-ES component during the search of a 2D latent space on a randomly chosen problem instance. We can first observe that even for a specific problem instance, there are several high-performing areas of interest which highlights the advantage of having multiple search components. Furthermore, it shows how the evolution strategy explores the space. The search variance is initially high to improve exploration until the search center moves into a high-performing area, whereupon the variance is gradually decreased to better exploit these promising strategies. We

provide additional plots and explanation in Appendix E.2 for other problem instances, demonstrating the spread of the specialised areas depending on the problem instance.

Lastly, it is worth noting that the adaptation mechanism of COMPASS (CMA-ES search) comes with negligible time cost (e.g. three orders of magnitude smaller than the time needed for the environment rollout), which is a strength compared to the costly backpropagation-based updates performed in EAS. We provide additional time analysis in Appendix L.

Table 2: Results of COMPASS and the baseline algorithms with instance augmentation for (a) TSP and (b) CVRP. We also report COMPASS with no augmentation (no aug.).

(a) TSP

| | Training distr. | | | Generalization | | | | | | | | |
| | $n=100$ | | | $n=125$ | | | $n=150$ | | | $n=200$ | | |
| Method | Obj. | Gap | Time | Obj. | Gap | Time | Obj. | Gap | Time | Obj. | Gap | Time |
|---|---|---|---|---|---|---|---|---|---|---|---|---|
| CVAE-Opt | - | 0.343% | 6D | 8.646 | 0.736% | 21H | 9.482 | 1.45% | 30H | - | - | - |
| SGBS | 7.769 | 0.058% | 9M | - | - | - | 9.367 | 0.220% | 8M | 10.753 | 0.619% | 14M |
| SGBS+EAS-Lay | 7.769 | 0.058% | 3H | - | - | - | 9.359 | 0.174% | 1H | 10.727 | 0.40% | 3H |
| POMO (sampling) | 7.767 | 0.026% | 2H | 8.594 | 0.128% | 20M | 9.376 | 0.321% | 32M | 10.916 | 2.14% | 70M |
| Poppy 16 | **7.765** | **0.002%** | 2H | **8.584** | **0.009%** | 20M | 9.351 | 0.141% | 32M | 10.802 | 1.08% | 70M |
| EAS | 7.768 | 0.039% | 7H | 8.590 | 0.082% | 66M | 9.360 | 0.175% | 138H | 10.724 | 0.350% | 206M |
| **COMPASS (aug.)** | **7.765** | **0.002%** | 2H | **8.584** | **0.009%** | 20M | **9.350** | **0.043%** | 32M | **10.723** | **0.337%** | 70M |
| COMPASS (no aug.) | 7.765 | 0.002% | 2H | 8.586 | 0.036% | 20M | 9.354 | 0.083% | 32M | 10.724 | 0.348% | 70M |

(b) CVRP

| | Training distr. | | | Generalization | | | | | | | | |
| | $n=100$ | | | $n=125$ | | | $n=150$ | | | $n=200$ | | |
| Method | Obj. | Gap | Time | Obj. | Gap | Time | Obj. | Gap | Time | Obj. | Gap | Time |
|---|---|---|---|---|---|---|---|---|---|---|---|---|
| CVAE-Opt | - | 1.36% | 11D | 17.87 | 2.08% | 36H | 19.84 | 3.24 % | 46H | - | - | - |
| SGBS | 15.66 | 0.01% | 10M | - | - | - | 19.43 | 1.08% | 4M | 22.57 | 2.59% | 9M |
| SGBS+EAS-Lay | **15.594** | **-0.36%** | 6H | - | - | - | **19.168** | **-0.27%** | 2H | **21.988** | **-0.05%** | 5H |
| POMO (sampling) | 15.67 | 0.18% | 4H | 17.56 | 0.33% | 43M | 19.43 | 1.08% | 1H | 23.24 | 5.64% | 100M |
| Poppy 32 | 15.62 | -0.14% | 4H | 17.49 | -0.10% | 42M | 19.32 | 0.50% | 1H | 22.94 | 4.27% | 100M |
| EAS | 15.62 | -0.21% | 13H | **17.462** | **-0.22%** | 2H | 19.213 | -0.037% | 5H | 22.162 | 0.73% | 7H |
| COMPASS (aug.) | 15.65 | -0.00% | 4H | 17.52 | 0.09% | 42M | 19.33 | 0.56% | 1H | 22.55 | 2.49% | 100M |
| **COMPASS (no aug.)** | **15.594** | **-0.36%** | 4H | 17.511 | 0.06% | 42M | 19.313 | 0.49% | 1H | 22.462 | 2.10% | 100M |

## 5 Conclusion

We present COMPASS, a novel approach to solving CO problems using RL. Our approach is motivated by the observation that active search is a key component to finding high-quality solutions to NP-hard problems. Finding one-shot solutions that are near-optimal is believed to be impossible. Instead, COMPASS is trained to create a distribution of diverse and specialized policies, conditioned on a structured latent space. This diversification is achieved by using an objective that specializes areas of the space on sub-distributions of problem instances. By navigating this latent space at inference time COMPASS is able to find the most performant policy for a given instance. Empirical results show that COMPASS achieves state-of-the-art performance on 9 out of 11 standard benchmark tasks across three distinct CO problems, TSP, CVRP and JSSP, outperforming prior RL methods based on either stochastic sampling or fine-tuning. We extend the canonical evaluation sets with instances that are procedurally transformed using mutation operators introduced in prior work. This additional set of tasks enables us to assess the generalization ability of COMPASS. We show that COMPASS is particularly robust to out-of-distribution instances, achieving superior performance in all 18 tasks considered. To better understand the benefits of our search procedure, we provide an empirical analysis of the latent space's structure, along with evidence of how it is explored at inference time. We show that, despite having no explicit regularization during training, the latent space exhibits clear regions of interest, and our search procedure is able to explore this space using an evolution strategy to produce high-performing policies. Overall, COMPASS proves to be performant, robust, and versatile on many types of CO problems and is able to provide solutions quickly at a reasonable computational cost.

**Limitations and future work.** The diversity of the policies contained in the latent space is closely linked to the specialisation that can be obtained from the training distribution, and hence potentially limited. We would like to inspect whether a broader range of policies could be obtained by using an additional unsupervised diversity reward, or by procedurally diversifying the distribution used. Another limitation of our method is the lack of structure in our latent space. Although we proved that it was enough to be successfully explored by an evolution strategy, we hypothesize that a better defined space could be searched through faster. We would like to inspect the use of regularization terms during the training phase to achieve this.

## Acknowledgements

Research supported with Cloud TPUs from Google's TPU Research Cloud (TRC). We thank anonymous reviewers for comments and helpful discussions that helped improve the paper.

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
