# OpenReview forum: "Combinatorial Optimization with Policy Adaptation using Latent Space Search"
_NeurIPS.cc/2023/Conference — NeurIPS 2023 poster_

### Official Review · Reviewer_GNoz · 2023-06-28

**Soundness:** 2 fair
**Presentation:** 2 fair
**Contribution:** 3 good
**Rating:** 5
**Confidence:** 4

**Summary:**

This paper presents an interesting CO agent model that conditions policy by latent vectors and finds latent vectors through CMA-ES.
In addition, considering latent vector in the learning process, this paper presents a training method that induces the agent to be specialized for various instances.

**Strengths:**

1. The idea of ​​using latent vectors to condition the policy and updating latent vectors via CMA-ES during inference is novel.

2. The training method with this inference method in mind is also interesting.

**Weaknesses:**

1. Inference time is not included in Table1. Since COMPASS is a method of continuously finding a better solution during inference, it is crucial to include the inference time in the experimental results.

2. It is difficult to view the experimental results of COMPASS as state-of-the-art results considering EAS (Hottung et al., 2022), SGBS+EAS (Choo et al., 2022) and DPDP [1] in CVRP experiments.

3. Overall, the description of the model structure, training method, and inference method lacks details.


***

[1] Kool, Wouter, et al. "Deep policy dynamic programming for vehicle routing problems." Integration of Constraint Programming, Artificial Intelligence, and Operations Research, 2022.

**Questions:**

1. Could you elaborate more on the conditioned decoder?

2. Since Equation (1) lacks a baseline, the learning stability is likely to be poorer compared to methods such as POMO and Puppy, which use a baseline in gradient calculation. Is there a particular reason for not using a baseline?

---

> ### Author Rebuttal · Authors · 2023-08-09
>
> We thank the reviewer for their insightful comments and positive feedback. We have updated the paper accordingly and hope our answers further clarify the aspects of the COMPASS framework and the training procedure.
>
> > W1: Inference time is not included in Table1. Since COMPASS is a method of continuously finding a better solution during inference, it is crucial to include the inference time in the experimental results.
>
> We agree that this is an important context and will therefore include the inference times of all methods and CO problems (previously reported in the Appendix Table 2) to Table 1 of the main text.
>
> > W2: It is difficult to view COMPASS as SOTA considering EAS (Hottung et al., 2022), SGBS+EAS (Choo et al., 2022) and DPDP (Kool et al., 2022) in CVRP experiments.
>
> Whilst EAS and SGBS+EAS do provide stronger adaptation to larger CVRP instances, we emphasize that this performance comes also with practical trade-offs and that the totality of all experiments strongly supports COMPASS as the leading method.
>
> Concretely,  EAS has orders of magnitude more adaptable parameters that must be re-trained (with non-negligible overhead and scalability challenges) on every considered instance - we refer the reviewer to our response to point W4 of reviewer ubWG for a more detailed discussion of the relative trade-offs of EAS.
>
> Moreover, on 9 out of 11 standard benchmarking tasks and all 18 generalization tasks, COMPASS outperforms the prior state-of-the-art approaches. With respect to the specific methods raised by the reviewer; (i) COMPASS outperforms the results presented by DPDP [1] on both TSP100 and CVRP100, (ii) COMPASS could, in principle, be combined with SGBS and EAS to further improve performance as the distribution of specialized policies encoded in the latent-space could still be used in conjunction with beam-search or fine-tuning at inference time.
>
> > W3: Overall, the description of the model structure, training method, and inference method lacks details.
>
> Details on these points are provided in the Appendices of the original submission, however we accept that they are important context and will ensure each is either moved into, or explicitly referenced in, our next revision of the main text. Specifically;
> - The model architecture for TSP and CVRP is fully described in Appendix A.5.  The JSSP model is described in Appendix A.12.
> - The training procedure is described in the Methods section of the main paper, which refers to Appendix A.7 for further details including pseudo-code for, and step-by-step details of, the algorithm.
> - The inference procedure is described in the Methods section of the main paper, which references Appendix A.8 for additional details and discussion of alternatives inference-time search protocols.
>
> > Q1: Could you elaborate more on the conditioned decoder?
>
> The decoder is conditioned on a vector sampled from a 16-dimension latent space. This latent vector is concatenated with the key, query, and value inputs of the multi-head attention decoder module. This conditioning allows us to create distinct policies while processing the same observation from the environment. Each latent vector corresponds to a unique policy, and thus, sampling the latent space to obtain vectors that our model can condition upon gives us an infinite set of policies.
>
> The conditioned decoder is described in Appendix A.5.  As discussed in our response to W3, we will ensure that the updated manuscript contains a brief description regarding the conditioned decoder in the methods section along with an explicit reference to Appendix A.5 for further details.
>
> > Q2: Since Equation (1) lacks a baseline, the learning stability is likely to be poorer compared to methods such as POMO and Poppy, which use a baseline in gradient calculation.
>
> We thank the reviewer for highlighting this oversight; in fact COMPASS does use a baseline in the gradient calculation; specifically it is the same baseline used in POMO and Poppy. We will updated Equation (1), which defines the gradient of the COMPASS objective, to the following:
>
> $$ \\nabla\_\\theta J\_{\\text{compass}} = \\mathbb{E}\_{\\rho \\sim \\mathcal{D}} \\mathbb{E}\_{z\_1, ..., z\_N \\sim \\mathcal{P}\_z} \\mathbb{E}\_{\\tau\_i \\sim \\pi\_\\theta(\\cdot |z\_i)} [\\nabla\_\\theta \\log \\pi\_\\theta(\\tau\_{i^\\star} | z\_{i^\\star})R\_{i^\\star} - \\mathcal{B}] $$
>
> where B is the baseline.
>
> ---
>
> [1] Kool et al. "Deep policy dynamic programming for vehicle routing problems." CPAIOR (2022).

---

> > ### Comment · Reviewer_GNoz · 2023-08-14
> >
> > I appreciate the authors providing detailed responses. I have two questions regarding Table 1 in the newly attached PDF.
> >
> > (1) It appears that POMO is POMO Sampling. If this is correct? I suggest using the term "POMO Sampling" or "POMO(sampling)" to reduce confusion with POMO Greedy, if the POMO in Table 1 refers to POMO Greedy, then the execution time is excessively long.
> >
> > (2) The authors included SGBS as a baseline, not SGBS+EAS from the SGBS paper. Since COMPASS iteratively finds solutions during inference time, I believe it would be more appropriate to include SGBS+EAS as a baseline rather than just SGBS. Is there a specific reason for choosing to include SGBS instead of SGBS+EAS as the baseline?

---

> > > ### Author Response · Authors · 2023-08-14
> > >
> > > We thank the reviewer for their comments and hope our answers provide further clarity.
> > >
> > > > (1) Clarification concerning POMO Sampling.
> > >
> > > We use POMO Sampling, and we will change it to “POMO (sampling)” for clarity.
> > >
> > > > (2) The authors included SGBS as a baseline, not SGBS+EAS from the SGBS paper. Since COMPASS iteratively finds solutions during inference time, I believe it would be more appropriate to include SGBS+EAS as a baseline rather than just SGBS. Is there a specific reason for choosing to include SGBS instead of SGBS+EAS as the baseline?
> > >
> > > EAS is an orthogonal approach to both COMPASS and SGBS, as during inference, both methods can be combined with EAS to finetune on instances. Since both COMPASS and SGBS follow from the POMO model architecture and at inference, employ a novel search method to find better quality solutions, we believe it is fair to compare the two approaches. However, we are happy to provide additional comparison to SGBS + EAS; detailed for TSP and CVRP in the below table (for instance sizes reported in [1]).
> > >
> > > Overall, the addition of SGBS + EAS improves the optimality gap with respect to EAS alone, and leaves the overall comparison to COMPASS unchanged. Concretely, COMPASS outperforms EAS and SGBS + EAS on TSP and in-distribution CVRP; whilst taking significantly less time.  As discussed in section 4.1 of the paper (and in answer to W4 - reviewer ubWG), the additional capacity of EAS allows stronger adaptation to larger out-of-distribution CVRP instances. We will include these extended results in our revised manuscript.
> > >
> > > As we cannot update the pdf file with our additional results, we report below the tour length, gap to optimality, and runtime for SGBS+EAS and COMPASS.
> > >
> > > ### TSP:
> > >
> > > | Method | 100  | 150  | 200  |
> > > |-------------|------|------|------|
> > > | SGBS+EAS | 7.767, 0.035% (3H) | 9.359, 0.136% (1H) | 10.727, 0.378% (3H) |
> > > | COMPASS (ours) | 7.765, 0.002% (2H) | 9.350, 0.043% (32M) | 10.723, 0.337% (70M) |
> > >
> > > ### CVRP:
> > >
> > > | Method | 100  | 150  | 200  |
> > > |-------------|------|------|------|
> > > | SGBS+EAS | 15.594, -0.36% (6H) | 19.168, -0.19% (2H) | 21.988, -0.07% (5H) |
> > > | COMPASS (ours) | 15.594, -0.36% (4H) | 19.313, 0.49% (1H) | 22.462, 2.10% (100M) |
> > >
> > > [1] Choo et al., Simulation-guided beam search for neural combinatorial optimization. NeurIPS (2022).

---

> > > > ### Comment · Reviewer_GNoz · 2023-08-15
> > > >
> > > > I appreciate the authors for their response to the additional questions. My concerns have been mostly addressed. I will raise my rating to 5.

---

### Official Review · Reviewer_ubWG · 2023-07-02

**Soundness:** 3 good
**Presentation:** 3 good
**Contribution:** 3 good
**Rating:** 6
**Confidence:** 5

**Summary:**

Building upon a pre-trained neural constructive model (such as POMO), this paper proposes COMPASS, which introduces the idea of learning a continuous latent search space to fine-tune the pre-trained POMO model parameter. The latent space allows for the sampling of a vector, which the pre-trained POMO model uses as a conditional vector to generate its own parameters. After training such latent space, continuous optimization algorithms (such as CAM-ES) can be utilized to search this space, in order to yield the most performant POMO model parameters for each test instance during inference. This allows for per-instance search during inference, while avoiding the need to retrain the deep model for each new test data (as is the case with active search in EAS). Experiments on benchmarks verify that COMPASS outperforms the state-of-the-art baselines.

**Strengths:**

- The concept of learning a latent search space for fine-tuning parameters of an NCO model is novel and could potentially impact the NCO community positively. However, it is important to acknowledge that the idea of learning a latent search space followed by using continuous optimizer to search within the space is not new, as evidenced by the CVAE-Opt method (ICLR’21).
- The authors conducted comprehensive experiments, supported by detailed tables, figures, and useful visualizations.
- COMPASS achieves state-of-the-art performance on benchmark TSP-100 and CVRP-100 instances.


**Weaknesses:**

- Although the authors provided reasons, I still think it would be useful to benchmark COMPASS against POMO and EAS equipped with data augmentation, to gain a complete understanding of COMPASS's advantages. That is to say, all baseline should be at their best settings. Furthermore, the recent method SGBS (simulation guided beam search, NeurIPS’22) is also overlooked.
- The method by which POMO is conditioned on the vector sampled from the latent space is unclear. Specifically, given a pre-trained NCO model, how should the user determine which parameters to condition on in practice?
- The literature review lacks comprehensiveness, missing important works like CVAE-Opt, SGBS, and other recent works.
- COMPASS's generalization to larger sizes (CVRP-200) appears less efficient than EAS.

**Questions:**

1. Does the run time include the time taken for CAM-ES? If so, what is the ratio of CAM-ES time to the total time?
2. In your comparison table, were all results obtained on your server? If so, could you provide the CPU/GPU model type of your serverfor readers to fully understand the efficiency?
3. How long does it take to train COMPASS and how many computational cost?
4. It would be interesting to investigate if COMPASS could enhance neural improvement heuristics like LIH, thereby yielding variable neighborhood search.

**Limitations:**

The revised paper should mention more limitations and future works.

---

> ### Author Rebuttal · Authors · 2023-08-09
>
> We thank the reviewer for their constructive comments.
>
> > W1: ... it would be useful to benchmark COMPASS against POMO and EAS with data augmentation. Furthermore, SGBS … is overlooked.
>
> We are happy to provide additional benchmarking to allow comparison to published results and validate our implementations. These are provided in the attached pdf (Table 1) and will be added to the Appendices of a revised manuscript. We believe not using augmentations for the results in the main text remains suitable, as discussed below.
>
> **Benchmarking: augmentations and SGBS** The key messages are unchanged from the previously reported results.
> - COMPASS outperforms all baselines with instance augmentation for all TSP instance sizes.
> - COMPASS is the leading method for in-distribution CVRP, with EAS providing stronger adaptation on larger instances (we note the adaption of EAS does not come without tradeoffs as discussed in our response to W4).
> - We also add SGBS [1] and CVAE-Opt [2]. COMPASS significantly outperforms both methods for all instance sizes of TSP and CVRP.
>
> **Augmentation-free results** We reaffirm the motivation behind running COMPASS and the baselines without instance augmentation.
> - Instance augmentation is a domain-specific trick which cannot be used for all problems (e.g. JSSP, Knapsack).
> - Baseline methods still benefit from a strong exploration as they use multiple starting points [3] and a substantial search budget.
> - Using augmentations complicates fair comparisons with inference-time search methods. Typically, 8 augmentations are used based on [3]. However, this number is arbitrary. More augmentations increase the number of samples from a specific policy, limiting the number of  adaptive steps for methods like EAS and COMPASS.
>
> > W2: How is POMO conditioned on the latent space… how to determine which parameters to condition on in practice?
>
> The decoder is conditioned on a vector sampled from a 16-dim latent space and is fully described in Appendix A.5. Specifically, the latent vector is concatenated with the key, query, and value inputs of the multi-head attention. For clarity we will add a brief description in the Methods section with reference to A.5 for further details.
>
> Whilst COMPASS is agnostic to the network architecture, we did not extensively explore alternative conditioning methods and leave this to future work.
>
> > W3: The literature review [is] missing important works like CVAE-Opt, SGBS and other recent works.
>
> Our literature review focused on RL construction methods, but we agree that a broader review can benefit future readers. Therefore, we will include discussions of CVAE-Opt and RL improvement methods. SGBS is already included in the Related Work (L. 90-91). We also provide CVAE-Opt and SGBS as additional baselines in the attached pdf (Table 1), which will also be included in the revised manuscript.
>
> > W4: COMPASS's generalization to larger sizes (CVRP-200) appears less efficient than EAS.
>
> EAS does adapt to the largest CVRP instances more effectively, however to achieve this it (1) adapts orders of magnitude more parameters (e.g. for CVRP200, EAS adapts 200*128=25.6k parameters compared to the 16-dim latent vector of COMPASS), (2) requires computationally expensive test-time training. Increasing the capacity of COMPASS (e.g. a larger latent space) can be explored in future work and we note that COMPASS and EAS are not mutually exclusive, and so could be combined.
>
> Despite these points, COMPASS outperforms EAS on in-distribution CVRP, all considered TSP and JSSP instance sizes and all the 18 generalization tasks (where instances are procedurally transformed to be out-of-distribution).
>
> > Q1: Does the run time include the time taken for CMA-ES? What is the ratio of CMA-ES time to the total time?
>
> Run times include CMA-ES steps, though this adaption makes a negligible contribution to the overall timings (in contrast explicit fine-tuning methods, e.g EAS). Details are provided in Table 2 of the attached pdf and will be included in a revised Appendix.
>
> > Q2: In Table 1, were all results obtained on your server? Could you provide the CPU/GPU model type…?
>
> COMPASS’ results and all baselines (POMO, POPPY and EAS) were computed by us using a v3-8 TPU. We used previously released checkpoints for these models to ensure consistency. We will update the main paper to include these details.
>
> > Q3: How long does it take to train COMPASS and how much computational cost?
>
> The final COMPASS models are trained until convergence, for each problem the training time and environment steps are: 4.5 days (110M steps) for TSP, 5.5 days (76.5M steps) for CVRP and 4.5 days (4.2M steps) for JSSP. These details will be added to the revised manuscript.
>
> > Q4: It would be interesting to investigate if COMPASS could enhance neural improvement heuristics like LIH, thereby yielding variable neighborhood search.
>
> We agree this is a promising future direction. In LIH, finding a diverse set of candidate solutions is also desired (see section V.C of [4]). As COMPASS is applicable to any pretrained model, instead of using the same policy stochastically several times (multi-run), or a selection of, likely similar, policies found during training (multi-policy), LIH could use COMPASS to promote diversity and subsequently improve performance.
>
> > The revised paper should mention more limitations and future works.
>
> Appendix A.14 discusses limitations and future work, however we accept this should be presented more prominently. In the revised text we will update the conclusion accordingly.
>
> ---
>
> [1] Choo et al., Simulation-guided beam search for neural combinatorial optimization. NeurIPS (2022).
>
> [2] Hottung et al., Learning a latent search space for routing problems using variational autoencoders. ICLR (2020).
>
> [3] Kwon et al., Pomo: Policy optimization with multiple optima for reinforcement learning. NeurIPS (2020).
>
> [4] Wu et al., Learning Improvement Heuristics for Solving Routing Problems. IEEE (2022).

---

> > ### Comment · Reviewer_ubWG · 2023-08-14
> > **Thanks for the response**
> >
> > I appreciate the authors for the detailed reply. I have two remaining points:
> >
> > 1. As pointed out by Reviewer GNoz as well, it would be beneficial if the authors could showcase the performance of COMPASS against the SGBS+EAS+augmentation (which is the current state-of-the-art).
> >
> > 2. I remain of the opinion that augmentation is a useful and effective technique to enhance constructive solvers (to escape local minima). Regarding the new results:
> > * Why does 'COMPASS (aug)' underperform compared to 'COMPASS (ours)' on CVRP?
> > * Why are computation times identical for both augmented and non-augmented versions?

---

> > > ### Author Response · Authors · 2023-08-14
> > >
> > > We thank the reviewer for their feedback and comments, and hope our answer provides further clarifications.
> > >
> > > > 1. As pointed out by Reviewer GNoz as well, it would be beneficial if the authors could showcase the performance of COMPASS against the SGBS+EAS+augmentation (which is the current state-of-the-art).
> > >
> > > We are happy to provide additional comparison to SGBS + EAS; detailed for TSP and CVRP in the below table which includes the tour length, gap to optimality, and runtime (for instance sizes reported in [1]). In general, the results leave the overall comparison to COMPASS unchanged.  COMPASS outperforms SGBS + EAS on TSP and in-distribution CVRP; whilst taking significantly less time. The additional capacity of EAS allows SGBS + EAS a stronger adaptation to larger out-of-distribution CVRP instances.
> > >
> > > These results will be included in our revised manuscript. Lastly, we would like to reaffirm that EAS is an orthogonal approach to COMPASS and SGBS, in particular, EAS could be added to COMPASS just like it is added to SGBS.
> > >
> > > ### TSP:
> > >
> > > | Method | 100  | 150  | 200  |
> > > |-------------|------|------|------|
> > > | SGBS+EAS | 7.767, 0.035% (3H) | 9.359, 0.136% (1H) | 10.727, 0.378% (3H) |
> > > | COMPASS (ours) | 7.765, 0.002% (2H) | 9.350, 0.043% (32M) | 10.723, 0.337% (70M) |
> > >
> > > ### CVRP:
> > >
> > > | Method | 100  | 150  | 200  |
> > > |-------------|------|------|------|
> > > | SGBS+EAS | 15.594, -0.36% (6H) | 19.168, -0.19% (2H) | 21.988, -0.07% (5H) |
> > > | COMPASS (ours) | 15.594, -0.36% (4H) | 19.313, 0.49% (1H) | 22.462, 2.10% (100M) |
> > >
> > > > 2.1 Why does 'COMPASS (aug)' underperform compared to 'COMPASS (ours)' on CVRP?
> > >
> > > To facilitate a fair comparison between COMPASS with and without augmentations we maintain a constant budget for the overall number of samples (see our answer to the question below); therefore with 8x augmentations the search procedure of COMPASS has 8 times fewer CMA-ES optimization steps.  The improvement of ‘COMPASS (ours)’ over ‘COMPASS (aug)’ highlights that our method performs better when dedicating additional steps to adapting the policy, rather than exploration via augmentations.
> > >
> > > > 2.2 Why are the computation times identical for both augmented and non-augmented versions?
> > >
> > > The computation budget of augmented and non-augmented COMPASS is fixed (i.e. both methods are allowed to generate the same number of candidate solutions).  This enables a direct comparison between these approaches that is consistent with existing literature in terms of budget use.
> > >
> > > [1] Choo et al., Simulation-guided beam search for neural combinatorial optimization. NeurIPS (2022).

---

> > > > ### Comment · Reviewer_ubWG · 2023-08-15
> > > > **Thanks for the reply**
> > > >
> > > > I appreciate the authors for the new results and new response. All of my concerns have been addressed to some extent and I am happy to keep my support for acceptance.

---

### Official Review · Reviewer_Qr8r · 2023-07-02

**Soundness:** 4 excellent
**Presentation:** 4 excellent
**Contribution:** 3 good
**Rating:** 6
**Confidence:** 4

**Summary:**

This paper proposes COMPASS, an RL-based training framework to learn a diversified neural solver for combinatorial optimization problems. This framework trains a conditioned neural network conditioned on a prior vector sampled from a fixed distribution. During the training phase, multiple priors are sampled, and only the parameters corresponding to the best prior are updated. During the inference phase, the evolution algorithm is employed to find the best prior for the current instance.

The authors test its method on TSP, CVRP, and JSSP. The experiment design is solid, and the results are promising.



**Strengths:**

1. The resources, e.g., memory and time, used in the inference time are shorter than the current SOTA from its low-dimensional prior.

2. The learned conditional neural solver has the potential to generalize to the out-of-training distribution instances. That is also somehow verified from the section 4.2 experiments.



**Weaknesses:**

1. One key challenge in the RL for CO area is how to train/generalize a model to the real big cases, e.g., TSP1000/10000. This framework is designed to improve upon another neural solver. However, it cannot solve the real big cases.

2. In the inference time, the priors are sampled and selected using the evolution algorithm. This is somehow like a search-related method. It is not clearly verified "the improvement comes from a good conditional neural solver or the strong evolution algorithm".

**Questions:**

1. Like mentioned in weakness (2), I am wondering what's the results of comparing COMPASS with COMPASS base neural solver + beam search/heuristic search. The traditional search can use fewer resources.

2. For the deep learning-based baselines, comparing time is insignificant. But I am wondering about the time cost of COMPASS. This can give me more confidence to evaluate how efficient COMPASS is. And to evaluate whether it can be used for real-world problems.

3. To my knowledge, TSP 100 and TSP 200 do not makes too much difference for a neural solver. I am wondering what's the results of testing on TSP 1000.

---

> ### Author Rebuttal · Authors · 2023-08-09
>
> We thank the reviewer for their constructive feedback and hope that our answers and additional experiments will clarify any concerns.
>
> > W1: Key challenge in RL for CO is to train/generalize a model to big cases, e.g. TSP1000/10000. This framework is designed to improve upon another neural solver [but] cannot solve big cases.
>
> We agree with the reviewer that scaling models to larger instances is an important challenge. However, whilst this is not a key focus of our work, we nonetheless believe that our method can contribute towards this goal. Concretely, two crucial aspects for tackling larger instance sizes are generalization and scalable architecture. Our method is SOTA for generalization (see Fig. 3), with scalable latent space adaptation that is independent of instance size). Furthermore, our method is architecture-agnostic.  Although we do not innovate on architecture, we can benefit from any new scalable architecture; e.g. nothing prevents us from using COMPASS on DIMES [1] and hence solve much larger instances, although this is beyond the scope of this work.
>
> > W2: It is not clearly verified whether the improvement [at inference time] comes from a good conditional neural solver or the strong evolution algorithm.
>
> Our experimental results provide evidence that both aspects – (1) a well-trained conditional neural solver and (2) an efficient search algorithm – are critical for strong performance. It can be seen in both the reported results in the paper and the attached pdf with additional experiments. We agree this is a crucial point and will therefore update the manuscript accordingly.
>
> **Previous results** Point (1) is illustrated by Fig. 4 (main) which shows high-performing regions for a given instance. Point (2) is demonstrated by Fig 9 (appendix) which shows the principled search method significantly outperforms random search. Additionally, we see that the random search outperforms POMO and Poppy, confirming that the latent space “contains” high-performing and diverse policies.
>
> **New Exp. 1** This is further illustrated in Fig. 1 (attached pdf) which compares two COMPASS models (fully- vs. under-trained) solving TSP150 instances with two search methods (CMA-ES and uniform sampling). The results demonstrate that:
> - Both search methods for fully trained model outperform those for under-trained model, showing the importance of our training procedure.
> - Uniform search on the fully-trained solver outperforms CMA-ES search on the under-trained model, showing that the search alone is not sufficient.
>
> **New Exp. 2** Fig. 2 (attached pdf) presents the evolution of the latent space during training on a TSP150 instance. It can be seen that initially the space is uniform (no specialized regions exist). However, as training progresses, high-performing regions emerge (shown in red) which indicates specialization of policies within the latent space, and we also see the improved performance of the best conditioned policy.
>
> > Q1: What are the results of comparing COMPASS with COMPASS base neural solver + beam search/heuristic search.
>
> This can be approximated by comparing with SGBS, a heuristic approach to improve the performance of POMO [2]. A naive application of this heuristic on COMPASS (sampling a random POMO policy with no latent space search) is equivalent to POMO+SGBS. We report the results of POMO+SGBS in Table 1 (attached pdf) and show that COMPASS outperforms POMO+SGBS on the whole benchmark. This validates that it is worth searching for a good latent condition with the budget rather than fixing a random policy and using a beam search. Nevertheless, there may be a trade-off between search in latent space and heuristic solution search, which we will mention in the updated manuscript.
>
> > Q2: What is the time cost of COMPASS?
>
> Time performance is reported in Table 2 (appendix), but we will update Table 1 (main) with the times. These results show that (i) COMPASS is as fast as POMO and POPPY (time cost of CMA-ES search is insignificant) (ii) COMPASS is significantly faster than EAS, e.g. 4x faster on CVRP 200.
>
> The adaptation mechanism of COMPASS comes with negligible time cost. We ran additional experiments to time the solution construction process of COMPASS. Those are reported in Table 2 of the attached rebuttal file and will be added to the Appendix. In a complete rollout of TSP100, the CMA-ES sampling and update takes 0.28 milliseconds, which is three orders of magnitude smaller than the time of the 99 decoding steps (298 ms).
>
> > Q3: To my knowledge, TSP100 and TSP200 do not make too much difference for a neural solver. I am wondering what the results of testing on TSP 1000 are.
>
> We believe that the difference between TSP100 and TSP200 is important, especially when considering generalization to larger instances as all checkpoints are trained on TSP100. Additionally, TSP 125, 150 and 200 are commonly used benchmark sets from the literature [3-5] and therefore valuable to report for consistency and to enable direct comparison.
>
> We agree scaling to larger instances is a prescient challenge which we discuss in the context of COMPASS in our response to W1.  As an initial examination on TSP1000, we run COMPASS, Poppy and POMO on the instances from [1], taking 7H each. COMPASS outperforms them by a significant margin (29.28 vs. 39.03 & 50.02). EAS-Emb is intractable on TSP1000, but EAS-Tab is reported in [1] and is largely outperformed as well (49.56 in 63.45H).
>
> ---
>
> [1] Qiu et al., DIMES: A Differentiable Meta Solver for Combinatorial Optimization Problems. NeurIPS (2022).
>
> [2] Choo et al., Simulation-guided beam search for neural combinatorial optimization. NeurIPS (2022).
>
> [3] Grinsztajn et al., Population-based reinforcement learning for combinatorial optimization. (2022).
>
> [4] Hottung et al., Efficient active search for combinatorial optimization problems. ICLR (2022).
>
> [5] Kwon et al., Pomo: Policy optimization with multiple optima for reinforcement learning. NeurIPS (2020).

---

> > ### Comment · Reviewer_Qr8r · 2023-08-12
> >
> > Thanks for the explanations and new experiments. All my questions are somehow answered.

---

### Official Review · Reviewer_YBD4 · 2023-07-06

**Soundness:** 3 good
**Presentation:** 3 good
**Contribution:** 2 fair
**Rating:** 6
**Confidence:** 4

**Summary:**

The paper proposes a neural combinatorial optimization approach that allows for an extensive search for high-quality solutions. The approach uses reinforcement learning to train a network to construct solutions for the traveling salesman problem (TSP), capacitated vehicle routing problem (CVRP), and job shop scheduling problem (JSSP). During training, the network is trained to parameterize a diverse set of policies that are conditioned on a continuous latent space. At test time, a guided search is performed using Covariance Matrix Adaptation (CMA-ES) to find regions in the continuous latent space that are associated with policies leading to high-quality solutions. The experiments indicate that the proposed method provides a competitive performance and is able to generalize well to instances that are different from those seen during training.

**Strengths:**

- The results show that the method offers a very good performance on all considered problems.
- The paper addresses an important problem (designing a neural combinatorial optimization approach that is able to perform an extensive search).
- The authors perform some interesting generalization experiments that go beyond changing only the instance size and also considers other shifts in the distribution.
- The authors will make the code of their method publicly available.
- Overall, the paper is well written and clearly organized.

**Weaknesses:**

- The proposed method is very similar to the method from [Hottung et al.] which also trains a neural network conditioned on a continuous latent space to construct diverse solutions to routing problems. At test time, both methods search the continuous latent space using continuous optimization methods (in this work CMA-ES, in Hottung et al. differential evolution.) The authors should make this clear in the paper and discuss the differences between the methods.
- In general, the paper omits discussing most of the existing works on learning extensive search methods (or improvement methods) for combinatorial optimization problems. Instead the authors conclude that “[..] the field has reached a point where methods [...] can hardly make significant improvements [over quickly generated solutions when] given a budget for additional computation.” (page 2) In fact, many approaches have been proposed that aim to exploit bigger computation budgets and that perform an extensive, guided search:
   - Xinyun Chen and Yuandong Tian. Learning to perform local rewriting for combinatorial optimization. Advances in Neural Information Processing Systems 32, 2019.
   - André Hottung and Kevin Tierney. Neural large neighborhood search for the capacitated vehicle routing problem. European Conference on Artificial Intelligence, pages 443–450, 2020.
   - Kim, Minsu, and Jinkyoo Park. "Learning collaborative policies to solve NP-hard routing problems." Advances in Neural Information Processing Systems 34 (2021): 10418-10430.
   - Ma, Yining, et al. "Learning to iteratively solve routing problems with dual-aspect collaborative transformer." Advances in Neural Information Processing Systems 34 (2021): 11096-11107.

- The reported performance of the baseline approaches is significantly worse than in their original works.
   - The validity of the experimental results is limited by the fact that the authors remove a core component from the considered baseline approaches. More precisely, the authors do not use instance augmentation (which considers 8 different augmentations of each test instance during the search). Using augmentations of an instance is an established way of increasing the exploration during the search, because neural network based construction methods tend to generate different solutions for each augmented version of an instance). While it can be argued that an augmentation mechanism is a “domain-specific trick” it also can be considered unfair to remove a component from a baseline approach that encourages exploration without replacing it with a different component that increases exploration. Most methods will perform worse if  a component that the developers considered as given when designing the method (even if it can be easily replaced by something else) is removed. Hence, I suggest that the authors also report results for the baselines with augmentation.
   - The authors reimplement the baseline approaches even though their code is publicly available. This has some pros and cons. On the plus side, implementing all approaches within the same code base allows a fair comparison of the runtime unaffected by implementation tricks or differences in the speed of the used framework. However, (to my surprise) the authors do not report any runtimes in the main paper. On the negative side, there is a risk that the implementation does not work as well as the original approach due small mistakes/misunderstandings. Currently, it is not possible to evaluate if the implementation of the authors matches the performance of the original code base because the authors do not report results with the augmentation mechanism being enabled. Hence, I again suggest that the authors also report results for the baselines with augmentation to demonstrate that their reimplementation is correct (in that case having a unified code base for all approaches could even be considered a strength of the paper).
- The authors should make it more clear if they used identical test instances to earlier work or if they generated new test instances (I hope the former, because generating new test instances makes a comparison to earlier works unnecessarily difficult).

[Hottung et al.] Hottung, André, Bhanu Bhandari, and Kevin Tierney. "Learning a latent search space for routing problems using variational autoencoders." International Conference on Learning Representations. 2020.

**Limitations:**

- The paper does not discuss limitations of the proposed method.

---

> ### Author Rebuttal · Authors · 2023-08-09
>
> We thank the reviewer for their comments and positive feedback. We will update the paper accordingly and have added additional experiments to help address their concerns.
>
> > W1: The proposed method is very similar to CVAE-Opt [1]. The authors should make it clear in the paper and discuss the differences.
>
> We agree that the work of [1], CVAE-Opt, merits discussion, however, there are significant differences both algorithmically and in terms of empirical performance between this work and COMPASS. Specifically, whilst [1] also trains a neural network conditioned on a continuous latent space and explores it at inference time:
> - Our method is entirely trained with reinforcement learning and hence, unlike [1],  does not require any direct supervision from pre-solved instances and is hence amenable to problems where no good prior solver is available.
> - CVAE-Opt has to additionally train a recurrent encoder of (instance, solution) pairs.  By contrast, COMPASS considers the latent space to encode a distribution of complementary policies and can be easily applied to pre-trained models (e.g. POMO).
> - Our method significantly outperforms the work of [1] whilst also having a significantly shorter runtime, as reported in Table 1 of the pdf file attached (will be added to Appendix A.1.1).
>
> Whilst our initial intention was to focus our Related Work section on RL construction methods, we will update it to cover a larger set of methods and ensure that [1] is discussed in any revised manuscript.
>
> > W2: The paper omits discussing most of the existing works on learning extensive search methods (or improvement methods). [And statement L. 41/42 is misleading].
>
> The sentence (L. 41/42) was referring to RL construction methods, however we accept that this was not sufficiently clear and will update the phrasing to make this explicit. Additionally, we will update our Related Work section to inform readers that there also exists a literature on improvement methods which propose approaches to exploit increased computation time. Finally, we will present the most relevant papers on improvement methods in a new section of the Appendix.
>
> Since the literature on construction methods is both extensive and more directly related to our work, we believe it is important to ensure this remains the key focus of the main manuscript (similarly to [3]) with extended context provided in the Appendices.
>
> > W3: The validity of the experimental results is limited by the fact that the authors (...) do not use instance augmentation. I (...) suggest also reporting results for the baselines with augmentation to demonstrate that their re-implementation is correct.
>
> We agree with the reviewer that reporting the results with problem augmentation is an important step to validate our implementations, and hence to strengthen our experimental results. Consequently, we have re-run the entire benchmark with instance augmentation.  The results will be added to the Appendix and are reported in the file attached to our rebuttal. Concerning the runtime, they were already reported in Table 2 of Appendix A.1.1 but we will add them to Table 1 (main paper).
>
> With regards to the correctness of our implementations; the reported POMO results are run in-house and match, or outperform, the published results in [2] where available.  We also independently verified that our re-implementation of EAS on TSP100 and CVRP100 with augmentations is consistent with the published results.
>
> We think that it is fair to run the benchmark without instance augmentation. As mentioned by the reviewer, those are domain specific tricks that cannot always be used (e.g. JSSP). It is hence crucial to develop approaches that do not rely on those for exploration. In addition, it is important to mention that the baselines still benefit from a strong exploration in our benchmark, as they all use the multiple starting positions introduced in [2].
>
> Finally, we note the use of augmentations makes fair comparison to methods designed for inference-time search challenging. In the literature, 8 augmentations are typically used for each problem instance following the protocol of [2]; however this choice is arbitrary and in practice any number could be used. Larger numbers allow more deterministic samples from a specific policy, but therefore also leave less of the inference budget for subsequent steps with updated policies when using adaptive methods such as EAS and COMPASS. Indeed, in these cases the optimal number of augmentations would have to be tuned as an additional hyper-parameter of the search.
>
> That being said, we agree that for continuity with previous work, it is valuable to run and report those results, which we did for the rebuttal. Whilst the overall ranking of different methods is nearly always unchanged, we can still extract interesting observations from those results. In particular (i) COMPASS remains state-of-the-art on all instance sizes for TSP (ii) on CVRP, COMPASS makes better use of its budget by exploring its latent space rather than relying on the instance augmentation. In particular, COMPASS with no augmentation outperforms all other methods with augmentation on CVRP 100.
>
> > W4: The authors should make it clearer if they used identical test instances to earlier work.
>
> We reuse identical test instances as the methods we compare to. We will update the paper to make it clearer.
>
> > The paper does not discuss limitations.
>
> We discuss limitations of our method in the Appendix (A.14). We nevertheless agree that limitations should be mentioned in the main paper: we will add a paragraph in the conclusion.
>
> ---
> [1] Hottung et al., Learning a latent search space for routing problems using variational autoencoders. ICLR (2020).
>
> [2] Kwon et al., Pomo: Policy optimization with multiple optima for reinforcement learning. Neurips (2020).
>
> [3] Kim et al., Sym-NCO: Leveraging Symmetricity for Neural Combinatorial Optimization. Neurips (2022).

---

> > ### Comment · Reviewer_YBD4 · 2023-08-15
> > **Reviewer response**
> >
> > Thank you for your response!
> >
> > > The sentence (L. 41/42) was referring to RL construction methods, however we accept that this was not sufficiently clear and will update the phrasing to make this explicit.
> >
> > Even for construction methods the statement is not completely true. For example, the recently proposed Poppy method can benefit from longer runtimes (e.g., 4 hours for 10,000 instances). Overall, I feel like you want to highlight a gap in the literature that is not really there.
> >
> > > Consequently, we have re-run the entire benchmark with instance augmentation.
> >
> > Thank you for conducting the additional experiments. For EAS you seem to only report the results from the EAS paper instead of those from your own JAX implementation. What is the reason for that? Overall, I am surprised by the long runtimes of EAS reported in the main paper. For CVRP100 EAS takes more than twice as long as POMO. In the EAS paper, EAS takes only 30% more time than POMO when sampling an identical number of instances.
> >
> > > The results will be added to the Appendix and are reported in the file attached to our rebuttal.
> >
> > Given that you get one additional page for the final version of the paper, I think it would be better to include results with and without augmentation in the main paper in one single table. This makes it easier for the reader to get a complete picture and to understand the impact of augmentation on the different methods. It is also more fair because results would then be reported for each method on their best setting (as pointed out by reviewer ubWG).
> >
> > > In the literature, 8 augmentations are typically used for each problem instance following the protocol of [2]; however this choice is arbitrary and in practice any number could be used.
> >
> > The literature usually uses 8 augmentations because this is the number of unique unit square transformations (see Table 1 in the POMO paper). While different augmentation techniques are possible, the established 8x time augmentation method is a very natural approach for 2d euclidean distance routing problems.

---

> > > ### Author Response · Authors · 2023-08-17
> > > **Additional comments [1/2]**
> > >
> > > We thank the reviewer for their additional comments.
> > >
> > > > **Even for construction methods the statement is not completely true. For example, the recently proposed Poppy method can benefit from longer runtimes (e.g., 4 hours for 10,000 instances).**
> > >
> > > We agree that L. 41/42 lacks nuance and propose a rephrasing. Our aim was to underscore the challenges faced by existing methods as detailed in L. 28-40. In essence:
> > >
> > > 1. RL construction methods (e.g. POMO and Poppy) strive to enhance solution quality from a few rollout episodes, primarily using simple bulk stochastic sampling methods as a “search” heuristic during tests.
> > > 2. Search-based methods, (e.g. EAS), have practical challenges like inference-time training costs.
> > > 3. Existing approaches typically separate training the one-shot inference policy from formulating an effective search procedure. Notably, addressing this third point drives the motivation behind COMPASS.
> > >
> > > Given that the preceding paragraph already communicates these challenges, we find L. 41/42 redundant and subjective. We suggest its removal and merging L. 43-45 with the end of the previous section to read: "...rather current approaches typically completely decouple both.  The absence of an efficient search strategy is even more detrimental when…".
> > >
> > > On the Reviewer's point about Poppy benefiting from extended runtimes, we agree it outperforms methods like POMO. This edge comes from its pre-training of a fixed but diverse set of policies.  However, it still predominantly depends on direct stochastic sampling to improve on its few-shot performance. Notably, in our experiments, we've used Poppy as a benchmark and observed that COMPASS's inference-time adaptation offers marked enhancements in both performance and generalization.
> > >
> > > > **For EAS you seem to only report the results from the EAS paper instead of those from your own JAX implementation. What is the reason for that? Overall, I am surprised by the long runtimes of EAS reported in the main paper. For CVRP100 EAS takes more than twice as long as POMO. In the EAS paper, EAS takes only 30% more time than POMO when sampling an identical number of instances.**
> > >
> > > Due to the time constraints of the rebuttal, we relied on published results. However, to clarify concerns about the accuracy of our re-implementation, we replicated results on TSP100 and CVRP100. Our findings align with those from [1]:
> > >
> > > | Method | TSP100  | CVRP100  |
> > > |-------------|------|------|
> > > | EAS (paper) | 7.769 | 15.63 |
> > > | EAS (ours) | 7.768 | 15.62 |
> > >
> > > The timings in the rebuttal were also taken from [1]. Notably, our EAS execution (using both our codebase and hardware) is about 20% faster than [1]. For instance, on TSP100, our POMO and EAS runtimes are 2 and 4 hours respectively, while [1] reports 5 hours for EAS. Although this difference doesn't change the overall conclusion that COMPASS is significantly more efficient, we plan to update all benchmarks using our EAS version for better comparison.
> > >
> > > While our EAS runtimes differ more significantly from POMO than in [1], this doesn't indicate inefficiency. EAS has two main steps at test-time: (1) rolling out trajectories to sample solutions, and (2) backpropagation for network weight updates. POMO only involves step (1). We suspect the variation in relative speeds between these steps comes from our use of JAX, instead of PyTorch as in [1]. Given that both our network and environment are written in pure JAX, we can compile the entire rollout into a single optimized operation, making sampling trajectories (step 1) much faster.
> > >
> > > Overall, we emphasize that best-efforts were made to optimize the implementations of all considered methods (e.g. POMO, Poppy,  EAS, COMPASS). Ultimately, our reported runtimes will come from a single codebase that will be open-sourced along with the paper.

---

> > > > ### Author Response · Authors · 2023-08-17
> > > > **Additional comments [2/2]**
> > > >
> > > > > **Given that you get one additional page for the final version of the paper, I think it would be better to include results with and without augmentation in the main paper in one single table.**
> > > >
> > > > We agree that a fair comparison between methods, with full context provided is essential and are happy to work with the reviewer to ensure this is the case.  We note that when considering all baselines with and without augmentations, the overall messages of the paper and relative comparison of COMPASS to our baselines remains qualitatively unchanged.
> > > >
> > > > Our perspective was that adding the results with augmentations could distract from the key contributions of the paper by adding a comparison between two inference modes for which these key messages are unchanged. We also note that (1) augmentations is not applicable to one of our CO problems (JSSP) and (2) optimizing the performance of COMPASS with augmentations would require a further investigation into the tradeoff associated with using more of the search budget for augmentations instead of adaptation steps. For these reasons, we felt that including the results with augmentations in the appendix whilst referring to both these and the motivation behind not using augmentations was the best compromise.
> > > >
> > > > Nonetheless, we are happy to add the results to the main table if the consensus amongst the reviewers is that this would be best, as these extended results support our overall message.
> > > >
> > > > > **The literature usually uses 8 augmentations because this is the number of unique unit square transformations (see Table 1 in the POMO paper). While different augmentation techniques are possible, the established 8x time augmentation method is a very natural approach for 2d euclidean distance routing problems.**
> > > >
> > > > We agree that using 8x augmentation is a natural approach but, as said by the Reviewer, alternative augmentation techniques are possible (for example; rotating the problem by arbitrary angles and re-scaling to the unit-square) and it is not the case that the standard 8x-augmentations are in any sense optimal. Moreover, since not all CO problems can be augmented, we aimed to develop a method that does not rely on domain-specific exploration tricks.

---

> > > > > ### Comment · Reviewer_YBD4 · 2023-08-21
> > > > >
> > > > > Thank you for your response. Almost all of my concerns have been addressed. Assuming that the promised changes will end up in the final version of the paper, I increase my final rating to 6.
> > > > >
> > > > > >  We note that when considering all baselines with and without augmentations, the overall messages of the paper and relative comparison of COMPASS to our baselines remains qualitatively unchanged.
> > > > >
> > > > > I still believe that showing all results in the main paper would provide the reader with a better understanding of the performance of the different methods. The message that your method offers state-of-the-art performance without the using augmentation remains indeed unchanged. In fact, clearly showing that your approach even outperforms other approaches that use augmentation would strengthen that message in my opinion.

---

### Author Rebuttal · Authors · 2023-08-09

We thank the reviewers for their positive comments, feedback and suggestions. In particular we are pleased to see the contributions of our work; both methodological and strong empirical performance, highlighted.

We respond to each question and concern in detail for each reviewer independently. However, in general, the recurring themes were requests for (1) extended results and discussions of additional solving methods; either using instance augmentation common for TSP and CVRP (Qr8r, ubWG) or additional models such as CVAE-Opt, SGBS, DPDP (YBD4, ubWG, ubWG), and (2) clarification of technical details and timings (YBD4, Qr8r, ubWG, GNoz).

We have provided additional results relating to point (1) in the attached pdf, namely Table 1.  We find that the overall messages of the paper, such as COMPASS remaining the most performant RL method is unchanged. In particular, COMPASS significantly outperforms the additional baselines requested.  For point (2) we have provided all additional information requested; with supporting figures in the attached pdf, clarifying our manuscript where details were missing, and including additional references to the relevant section of the Appendices.

We thank the reviewers for raising these points as the additional results further strengthen our contributions.  We believe that all raised points have been addressed and would be happy to discuss any remaining concerns that the reviewers may have.

---

### Decision · Program_Chairs · 2023-09-21

**Decision:**

Accept (poster)

**Comment:**

The paper proposes a neural combinatorial optimization approach to produce high-quality solutions. The approach employs reinforcement learning to train a network to parameterize a diverse set of policies that are conditioned on a continuous latent space. For a new testing problem, guided search is performed using CMA-ES to find regions in the latent space that are associated with policies leading to high-quality solutions. Experiments show good results on benchmark combinatorial optimization problems.

Overall, reviewers' were positive about the paper and author rebuttal addressed their questions and concerns.

I recommend accepting the paper and strongly encourage the authors' to incorporate reviewers' comments and rebuttal discussion into the final paper.